# O/W Pickering Emulsions Stabilized with Cellulose Nanofibrils Produced through Different Mechanical Treatments

**DOI:** 10.3390/foods10081886

**Published:** 2021-08-15

**Authors:** Annachiara Pirozzi, Roberta Capuano, Roberto Avolio, Gennaro Gentile, Giovanna Ferrari, Francesco Donsì

**Affiliations:** 1Department of Industrial Engineering, University of Salerno, via Giovanni Paolo II, 132, 84084 Fisciano, SA, Italy; apirozzi@unisa.it (A.P.); gferrari@unisa.it (G.F.); 2Institute for Polymers Composites and Biomaterials, National Research Council of Italy, Via Campi Flegrei 34, 80078 Pozzuoli, NA, Italy; roberta.capuano@ipcb.cnr.it (R.C.); roberto.avolio@ipcb.cnr.it (R.A.); gennaro.gentile@ipcb.cnr.it (G.G.); 3ProdAl scarl, University of Salerno, via Giovanni Paolo II, 132, 84084 Fisciano, SA, Italy

**Keywords:** nanocellulose, emulsion stability, high-pressure homogenization, ball milling, interfacial tension

## Abstract

This work aimed at studying the stabilization of O/W Pickering emulsions using nanosized cellulosic material, produced from raw cellulose or tomato pomace through different mechanical treatments, such as ball milling (BM) and high-pressure homogenization (HPH). The cellulose nanofibrils obtained via HPH, which exhibited longer fibers with higher flexibility than those obtained via ball milling, are characterized by lower interfacial tension values and higher viscosity, as well as better emulsion stabilization capability. Emulsion stability tests, carried out at 4 °C for 28 d or under centrifugation at different pH values (2.0, 7.0, and 12.0), revealed that HPH-treated cellulose limited the occurrence of coalescence phenomena and significantly slowed down gravitational separation in comparison with BM-treated cellulose. HPH-treated cellulose was responsible for the formation of a 3D network structure in the continuous phase, entrapping the oil droplets also due to the affinity with the cellulose nanofibrils, whereas BM-treated cellulose produced fibers with a more compact structure, which did adequately cover the oil droplets. HPH-treated tomato pomace gave similar results in terms of particle morphology and interfacial tension, and slightly lower emulsion stabilization capability than HPH-treated cellulose, suggesting that the used mechanical disruption process does not require cellulose isolation for its efficient defibrillation.

## 1. Introduction

Emulsions are mixtures of two immiscible liquids, with one of them finely dispersed in the form of small droplets into the continuous phase of the other liquid [1], for example, oil droplets dispersed in water (O/W emulsions). Since emulsions are thermodynamically unstable systems, tending toward complete phase separation, their properties are bound to change over time. The rate of change of emulsion properties, and, in particular, droplet size distribution, in comparison with the expected shelf life of the product containing the emulsion, define emulsion kinetic stability [2]. Different mechanisms accelerate emulsion instability phenomena. Physical instability mechanisms include flocculation, coalescence, Ostwald ripening, gravitational separation, and phase inversion [3], whereas the most common sources of chemical instability are lipid oxidation and degradation of functional ingredients, such as aroma, flavor, or health-beneficial compounds [4]. A (kinetically) stable emulsion can resist the environmental stimuli experienced during emulsion incorporation into the final product and subsequent transformation, storage, and preparation, such as exposure to extremes of temperature, pH, ion concentration, radiation, or shearing [3]. Optimal emulsion performance, in terms of ease of preparation and stability over time, is ensured by the proper selection of interfacial agents, such as surfactants or polymeric emulsifiers, which reduce the interfacial tension between the two phases, stabilizing the emulsion droplet during emulsification and storage and therefore delaying phase separation.

Emulsions can be efficiently stabilized also by colloidal particles, instead of surfactant or emulsifier molecules, forming the so-called Pickering emulsions [3]. Solid particles irreversibly adsorb at the liquid/liquid interfaces because of their high interfacial adsorption energy, forming a dense film around the oil droplets, which prevents droplet coalescence mainly through steric repulsion [5], maintaining excellent stability over time regardless of the external environmental stimuli. Moreover, since several edible natural substances (e.g., polysaccharides, proteins, and lipids) can be used as solid particulate stabilizing materials, Pickering emulsions generally exhibit lower toxicity and better biocompatibility and food compatibility than emulsions stabilized by surfactants [1,2,6,7]. Many researchers have focused on the utilization of proteins for this purpose, derived from animal, plant, or microbial sources [8]. Although protein particles can be produced in nanometric size, protein-based particle-stabilized emulsions are generally sensitive to coalescence at pH around their isoelectric points and are susceptible to proteolytic enzymes [9]. Recently, polysaccharide-based particles have attracted increasing attention as potential stabilizers in multi-phase food systems. Among polysaccharides, nanostructured cellulose is particularly suitable for the stabilization of edible Pickering emulsions, of interest in the fields of biomedicine, food, and cosmetics [6].

As alternatives to synthetic particles, cellulose emerged as a potential and novel stabilizer of food-grade Pickering emulsion, due to its sustainability, biodegradability, and nontoxicity. More specifically, Pickering emulsions stabilized by cellulose particles have attracted increasing interest in different fields, because they confer noteworthy stability against coalescence, adjustable permeability, and good elastic responses [10,11]. However, it is still not clear the role of processing conditions on the techno-functional properties of cellulose for the stabilization of Pickering emulsions. It was demonstrated that nanosized cellulosic (NC) materials possess several advantages deriving from their nanometric sizes, such as high tensile strength, stiffness, and surface functional groups. When used as a stabilizer in O/W emulsions, NC is reported to significantly enhance the properties and performance in comparison with conventional systems [12].

NC is generally prepared using a top-down approach to isolate the semi-crystalline individual nanofibers or extract the crystalline portion. However, the chosen process and its processing conditions determine the type of nanocellulose produced: cellulose nanocrystals (CNCs) are produced from chemical and enzymatic treatments, and cellulose nanofibrils (CNFs) are obtained by either mechanical, such as high-pressure homogenization (HPH), or chemical treatments [13,14,15]. In particular, HPH is reported to induce the complete activation (or fibrillation) of cellulose fibers, through the formation of CNFs, characterized by favorable surface-to-volume ratios and mechanical strength [16]. In a different approach, cellulose fibers can be destructured by a dry mechanical process, such as ball milling (BM), obtaining amorphous cellulose particles (CPs) [17]. These amorphized cellulose particles, when used as filler in polymer composites, have shown good dispersibility in different polymer matrices [18,19].

The present study aimed at comparing, for the first time, two different mechanical treatments, such BM or HPH, in terms of their effect on the properties of cellulose fibers for the stabilization of Pickering O/W emulsions under different environmental conditions. Moreover, this work also aimed at extending the most promising mechanical treatment to an agri-food residue rich in cellulose, such as tomato pomace, for assessing also the use of an unrefined cellulose source in the stabilization of Pickering emulsions.

## 2. Materials and Methods

### 2.1. Materials

Commercial cellulose Arbocel^®^ BWW40, with a declared average length of the cellulose fibers of 200 µm, was kindly supplied by Rettenmaier Italia (Brescia, Italy). Tomato pomace was kindly supplied by Salvati Mario & C. spa (Mercato San Severino, Italy). Peanut oil was bought from a local supermarket (Giglio Oro, Carapelli Firenze Spa, Firenze, Italy). Methylparaben, Nile red (purity ≥ 98%), NaOH, HCl, and NaCl were purchased from Sigma-Aldrich—Merck KGaA (Darmstadt, Germany). All reagents were of analytical grade unless otherwise stated. Micro-pure deionized water was used throughout this work.

### 2.2. Mechanical Treatments of Cellulose and Tomato Pomace

Ball-milled cellulose particles (CP) and cellulose nanofibrils (CNF) were produced from commercial cellulose fibers through ball milling (BM) and high-pressure homogenization (HPH), respectively. For BM, cellulose was preliminarily dried at 90 °C under vacuum for 24 h, and then samples (10.0 ± 0.1 g) were treated in a Retsch PM100 planetary ball-milling system (Haan, Germany), using a 125 mL steel milling cup and 25 steel spheres with a diameter of 10 mm. The weight ratio of spheres/cellulose was about 10:1. The ball-milling process was carried out for 30 min (BM30-CP) or 60 min (BM60-CP).

The HPH treatment, which was tested for its known ability to defibrillate cellulose [20], required cellulose dispersion in water at 5 g/L (optimized concentration for emulsion preparation, resulting from preliminary trials). Samples (200 mL) were pretreated using a shear mixer (Ultra Turrax T-25, IKA Labortechnik, Staufen, Germany) at 20,000 rpm for 5 min, within an ice bath, before HPH treatment at 80 MPa for 10 min at a flow rate of 22 mL/s (HPH-NC), using an in-house developed unit, equipped with a 200 µm diameter orifice valve (model WS1973, Maximator JET GmbH, Schweinfurt, Germany). The use of smaller orifice valves (and, hence, of higher operating pressures) was not possible because of valve clogging. Tube-in-tube heat exchangers, with cold water at 5 °C, were placed immediately upstream and downstream of the orifice valve to ensure that the product temperature was always kept < 10 °C.

Additionally, to prove the concept of the use of an unrefined source of cellulose for emulsification instead of NC from pure cellulose, the most promising treatment (HPH) has been also applied to a tomato pomace suspension. The tomato pomace (moisture 80.70 ± 0.83 g/100 g), milled in a laboratory blender, was then mixed with water to a final concentration of 5 g/L of dry pomace. The obtained suspension was treated with the shear mixer at 20,000 rpm for 5 min in an ice bath, and, subsequently, by HPH (orifice diameter of 200 μm) at 80 MPa and 25 °C for 10 min at a flow rate of 22 mL/s (sample HPH-TP). A higher processing temperature (25 °C instead of 5 °C) was adopted to improve the processability of the tomato peels [21].

### 2.3. Preparation of Pickering Emulsions

BM cellulose was dispersed at 5 g/L of CP (concentration selected through preliminary trials for the Pickering emulsions) into distilled water at room temperature (25 ± 1 °C), to reach the same solid concentration of HPH-NC and HPH-TP suspensions, to form the aqueous phase. Methylparaben was preliminarily dissolved in hot peanut oil (70 °C) at a concentration of 2 g/kg under stirring for 5 min using a laboratory hot plate/stirrer. Then, 100 g/kg of oil phase was mixed with the aqueous phase in a shear mixer at 20,000 rpm for 5 min in an ice bath, to obtain a coarse O/W emulsion. Subsequently, the coarse emulsions were treated through HPH at 80 MPa for 15 min at a flow rate of 22 mL/s, with the heat exchangers set at 5 °C.

### 2.4. Light Microscopy Analysis

The microscopic structure of samples was observed using an optical inverted microscope Nikon Eclipse (TE 2000S, Nikon instruments Europe B.V., Amsterdam, The Netherlands) equipped with a polarization filter, with a 10× objective, coupled to a DS Camera Control Unit (DS-5M-L1, Nikon Instruments Europe B.V, Amsterdam, The Netherlands) for image acquisition and analysis. Water suspensions (10 µL) of the different samples were poured on a microscope slide and covered with a cover glass. For the visual observation, the light microscope with the photographing option in the DS Camera Control Unit was used. Prepared solutions were observed also using a polarizing light to observe NCs birefringence.

The anisotropic properties and the microstructure of the different Pickering emulsions were also measured with the same microscope. For fluorescence measurements, 10 μL of Nile Red (1 mg/mL in ethanol) was added to 100 μL of collected samples to stain the oil phase before observations.

### 2.5. Scanning Electron Microscopy Analysis

Scanning electron microscopy (SEM) analysis was performed on neat cellulose, BM, and HPH samples previously dried at room temperature. The analysis was performed with an FEI Quanta 200 FEG SEM using a secondary electron detector and an acceleration voltage of 10–30 kV. Before the SEM imaging, the samples were sputter-coated with a 10 nm thick gold/palladium layer.

### 2.6. Interfacial Tension Measurements

The interfacial tension at O/W interfaces was measured for the different aqueous suspensions (cellulose, BM30-CP, BM60-CP, HPH-NC, and HPH-TP) using a pendant drop tensiometer (KSV Instruments LTD CAM 200, Helsinki, Finland), equipped with an image analysis software. Each suspension at 5 g/L concentration on a dry basis was placed in a 500 µL syringe (Hamilton Company, Bonaduz, Switzerland) equipped with a 0.72 mm diameter stainless steel needle, and forming a drop of about 2 µL in a quartz cuvette (3 mL) filled with the oil phase. The peanut oil was previously filtered with a PTFE syringe filter (pore size 0.45 µm) to remove some of the impurities, which can negatively influence the interfacial tension measurements. All tests were carried out at 25 °C. The interfacial tension (γ) measurements were performed each 10 s, over 500 s.

### 2.7. Size Distribution of BM-CP, NC, HPH-TP, and Emulsions

The size distributions of aqueous suspensions and emulsions were measured by laser diffraction using a Mastersizer 2000 instrument (Malvern instrument Ltd., Malvern, UK), using the Fraunhofer approximation, which does not require knowledge of the optical properties of the sample. The temperature of the cell was maintained at 25 ± 0.5 °C, and an average of triplicates was determined. The surface-weighted mean diameter D[3,2] and volume-weighted mean diameter D[4,3] were evaluated according to Equations (1) and (2), respectively:(1)$$D\left[3,2\right]=\frac{\sum_{i}n_{i}d_{i}^{3}}{\sum_{i}n_{i}d_{i}^{2}}$$
(2)$$D\left[4,3\right]=\frac{\sum_{i}n_{i}d_{i}^{4}}{\sum_{i}n_{i}d_{i}^{3}}$$
where *n_i_* is the number of particles of diameter *d_i_*.

Additionally, d(0.1), d(0.5), and d(0.9) were evaluated, corresponding to the 10th, 50th (median value), and 90th percentile of the cumulative size distribution of the suspensions. The relative span factors were calculated following Equation (3), to express the distribution width of the droplet size distribution:(3)$$\mathrm{Span}=\frac{d\left(0.9\right)-d\left(0.1\right)}{d\left(0.5\right)}$$

### 2.8. Emulsion Rheological Analysis

Rheological measurements of emulsions were performed in a rotational rheometer (AR 2000 rheometer, TA instruments, Newcastle, DE, USA), equipped with a plate-cone geometry (40 mm diameter and 2° of cone angle) with a fixed gap width of 1 mm at 20, 50, and 80 °C, in a stationary regime. The apparent viscosity of the emulsions was determined as a function of shear rate, as previously reported for the characterization of Pickering emulsions stabilized with cellulose microfibers [22]. Flow curves were obtained by continuously varying the shear rate from 0.1 to 300 s^−1^.

### 2.9. Influence of Environmental Factors on Emulsions Stability

Emulsion stability was assessed by visual observation and monitored by taking photos for a storage period of at least one month, at 4 °C. Moreover, accelerated stability tests against gravitational separation were also conducted, by centrifuging the emulsions at 3500 rpm for 10 and 20 min, which are considered equivalent to a gravitational acceleration field applied for around 6 and 12 months, respectively [23]. The effects on the stability of the Pickering emulsions of pH over storage were investigated through microstructure observation and rheological analysis. The pH of the emulsions was adjusted to the values of 2.0, 7.0, and 12.0, which were selected as representative of neutral pH and two pH extremes, using a 0.1 mol/L of HCl solution and a 0.1 mol/L of NaOH solution. The pH values were controlled through direct measurement in the emulsions.

### 2.10. Statistical Analysis

Statistically significant differences (*p* ≤ 0.05) between the means were evaluated using a one-way analysis of variance (ANOVA), performed with SPSS 20 (SPSS Inc., Chicago, IL, USA) statistical package, and Tukey’s test. At least three replicates were conducted for each analysis otherwise specified. All the data were reported in the form of mean ± SD.

## 3. Results and Discussion

### 3.1. Structural Analysis of CP, CNF, and HPH-TP

#### 3.1.1. Morphology Analysis

The morphological and structural parameters of cellulose structures obtained via mechanical treatments and of HPH-treated tomato pomace were investigated through optical microscopy (Figure 1) and SEM analysis (Figure 2). The mechanical processes caused irreversible alterations in the cellulosic fibers, the capability of mutual interaction, and fibers entanglement. In the case of tomato pomace, HPH caused the cell disruption into filamentous debris appearing in the suspension. Native cellulose (Figure 1a) exhibited a fibrous structure and the typical birefringence under polarized light. BM cellulose (at 30 and 60 min) significantly differed in fiber length from native cellulose. Ball-milling treatments, due to the efficient disintegration of long cellulose chains, facilitated the production of BM-CPs characterized by a relatively constant cross-section and high birefringence properties (Figure 1b,c). In contrast, HPH-treated cellulose (Figure 1d) exhibited a less ordered and uniform structure. The micrometric cellulose particles that can be observed after mechanical treatments can be attributed to the residual aggregated fibril bundles after partially removing the cellulose amorphous region [24]. These bundles are characterized by a large variation in cross-section (irregular widths), and typical lengths in the range of ∼10–100 μm. The observation that the birefringence property observed under polarized light did not significantly change with the HPH treatments suggests that mechanical processes are likely to cause the structural disruption of cellulose fibers, without affecting the crystalline organization of the treated cellulose fibers.

In the case of tomato pomace, the HPH process caused the rupture of cell walls and membranes, due to the extremely intense fluid-mechanical stresses (shear, elongation, turbulence, and cavitation) generated in the homogenization valves, reducing the size of the suspended material and liberating all the intracellular compounds, as shown by microscopy analysis, already in previous work [21], and in detail, for different HPH treatment times in the Appendix A. Moreover, polarized light microscopy of the HPH-TP sample showed its highly birefringent properties, which can be attributed to the crystalline region resulting from the orderly arrangement of cellulose molecules in microfibrils in the cell wall [25].

SEM analysis confirmed the effect of the processes on the morphology of the samples. Indeed, SEM images reported in Figure 2a,b revealed the fibrous structure of untreated cellulose, with an approximate diameter of about 20 µm and an approximate length of 200 µm. The BM process induced a progressive disruption of the cellulose fibrous structure, with the formation of well-separated cellulose particles with an irregular shape and a lateral size in the range of 10–50 µm, slightly lower in the case of cellulose treated for 60 min (Figure 2c–e). At higher magnifications (Figure 2d–f), the BM cellulose showed a compact morphology that did not evidence the presence of typical cellulose fibrils. In stark contrast, the HPH processes of cellulose and tomato pomace aqueous suspensions induced the obtainment of a cellulose-based material that, after air drying at room temperature, produced the formation of a compact film (Figure 2g–i). For the HPH-treated cellulose, the original structure of the cellulose fibers is still evident in Figure 2g, whereas the drying of the HPH-treated tomato pomace suspension produced a highly smooth film with micrometric-sized pores (Figure 2i). For the HPH-treated materials, the good film-forming ability can be ascribed to the pronounced defibrillation of the original structures, clearly evidenced in Figure 2h–j and the establishment of strong interactions between the nanosized cellulose fibrils during the drying process.

#### 3.1.2. Interfacial Tension

The effect of the suspended particles, processed with different mechanical treatments, on the interfacial tension at the O/W interface, is a predictor for emulsion formation and stability, as well as its mean droplet size [26]. In general, the higher the interfacial tension, the more resistant the whole system is to deformation and the more likely phase separation between the two liquids occurs. 

The results reported in Figure 3, as kinetic data of interfacial tension, clearly show that the oil phase still contains significant impurities, also after filtering the oil with a PTFE filter. Therefore, the interfacial behavior of the samples is dominated by the behavior of the control sample. In general, the systems where cellulose was added did not show any noteworthy reduction in interfacial tension compared to the control, with the exception of the HPH-NC sample, which exhibited an interfacial tension that was consistently lower than the other samples. More specifically, the BM30-CP and BM60-CP samples did not show an enhanced surface activity compared to the untreated cellulose, with the dynamic interfacial tension curves for cellulose and BM30-CP that are almost identical and the BM60-CP curve being slightly higher. The observation that the dynamic curves for cellulose suspensions do not differ from the curves for pure water in oil is consistent with previous data showing that cellulose particles are not particularly surface active in the absence of salt [27,28]. 

The lower interfacial tension dynamic curves observed for HPH-NC could be likely ascribed to the morphology changes induced by HPH, promoting defibrillation but less efficient fiber breakage than BM. Moreover, it can be hypothesized that HPH treatment caused the cellulose to exhibit a lower interfacial tension than cellulose, due to the stronger attractive capillary forces establishing for the softer particles generated by HPH treatment than for rigid ones [29]. In the case of tomato pomace, its complex composition and reduced cellulose content might explain the reduced differences observed with the control sample, despite the HPH treatment. 

Based on the interfacial tension data, it can be expected that HPH-NC could be more suitable for the stabilization of emulsions [30] than the other treated cellulose samples. As a final remark, it must be highlighted that the decrease in interfacial tension induced by the suspended cellulose particles was of a rather small entity in comparison with the interfacial tension values reached by highly surface-active molecules such as surfactants and proteins [31,32].

#### 3.1.3. Effect of Mechanical Treatments on the Particle Size Distribution

Table 1 reports the particle size distribution of pure cellulose, BM and HPH-treated cellulose, and HPH-treated tomato pomace in terms of d(0.1), d(0.5), and d(0.9), as well as span and D[4,3] and D[3,2]. Remarkably, the particle size of cellulose was significantly reduced after ball-milling treatment (D[4,3] decreased by ~55%, D[3,2] of ~45–50%), but longer BM treatment times did not significantly change the size distribution. This is evident also when considering the diameters corresponding to the 10th, 50th, and 90th percentiles, with statistically significant differences (*p* < 0.05) observed only for d(0.9).

The HPH treatment also caused a significant reduction in cellulose particle size (D[4,3] decreased of ~45%, D[3,2] of ~28%). Interestingly, the surface mean diameter D[3,2] decreased less than the volume mean diameter D[4,3] and was significantly higher than for BM samples, suggesting that less compact fibers, characterized by a higher surface area, are formed by HPH, in agreement with optical microscopy and SEM observations. Moreover, it can also be observed that the HPH treatment is less efficient for cellulose micronization than BM, with generally higher values of the diameters corresponding to the 10th, 50th, and 90th percentiles. This is confirmed also by the span values: for BM-treated cellulose, a minimum span of the particle size distribution was observed (values of 2.58 and 2.48, respectively, after 30 and 60 min of BM treatment), whereas, for HPH, a span value of 3.10 was achieved (for comparison, the span in raw cellulose was 3.61). In the case of tomato pomace, because of the reduced fiber content (about 34% on a dry basis), a narrower particle size distribution was obtained through HPH treatment, with a span of the distribution of 2.35.

In the case of tomato pomace, a direct comparison with cellulose is not meaningful, as the starting material is completely different, and components other than fibers (such as proteins, lipids, and polysaccharides) are present. However, the size distribution data, with the help of the microscopic images (see Figure 1 and Figure 2), suggest that HPH induced not only almost complete cell disruption of tomato pomace but also cellulose defibrillation. Further studies are required to better clarify the aspects related to processing unrefined biomass for cellulose defibrillation.

### 3.2. Characterization of Nanocellulose-Stabilized Pickering Emulsions

The optical microscopic images and digital pictures of Pickering emulsions stabilized with cellulose treated with different mechanical methods and with HPH-TP are shown in Figure 4. In the right column of Figure 4, cellulose and oil are distinguishable in the fluorescence micrographs, where the cellulose birefringence can be recognized by yellowish shades, and the Nile-red stained oil is also evident as red domains. The appearance of the emulsions prepared with BM cellulose, shown in Figure 4a,b, suggests that the cellulose particles do not completely cover the oil droplets, but remain largely dispersed in the aqueous phase. This effect is also confirmed by the fluorescence micrographs in Figure 4a,b (corresponding to the brightfield micrographs in the left column), which highlights the uniform fiber dispersion in water as a green shade observed in the entire continuous phase. This agrees well with the limited stability of these emulsions, associated with droplet coalescence, causing complete phase separation within 24 h after preparation (data not shown). Interestingly, for the emulsions prepared with HPH-NC and HPH-TP (Figure 4c,d), the particles form large irregular domains aggregated around the oil droplets, efficiently entrapping them and maintaining the oil in emulsified form. In the case of HPH-NC and HPH-TP emulsions, no fluorescent reflection was observed in the continuous phase. This observation indicates that these solid particles are mainly aggregated around the oil droplets and close to the oil–water interface, leading to the efficient stabilization of the Pickering emulsions. In these emulsions, no coalescence phenomena were observed over 28 d at 4 °C (no evidence of oiling off), except gravitational separation and associated creaming. Vigorous mixing was sufficient to form again the emulsion. These systems displayed outstanding stability against droplet coalescence likely as a consequence of steric stabilization because of the different cellulose configuration in comparison with BM cellulose, and also possibly because of the thickening effect of the aqueous phase due to the longer cellulose chains [33].

The size distributions of the different Pickering emulsions are presented in Figure 5. Remarkably, it can be observed that they are coherent with the micrographs of Figure 4. In the case of the emulsions prepared with HPH-treated cellulose, a main peak can be observed around ~20 μm, corresponding to the cellulose-rich aggregates around the oil droplets, and a distinct minor peak around ~1 μm, which can be likely attributed to individual cellulose fibrils dispersed in the aqueous phase. A similar distribution occurs for tomato pomace, but with the main peak centered around ~50 μm. In the case of BM-treated cellulose, a wider distribution can be observed for both samples (BM30-CP and BM60-CP emulsions), corresponding to the observation that the cellulose particles remain dispersed in the aqueous phase, rather than covering the oil droplets. These considerations are supported by the values of the characteristic diameters of the distributions and span values reported in Table 2. Remarkably, Table 2 shows smaller span values for HPH-treated cellulose or tomato pomace emulsions than for BM-treated cellulose emulsions, despite the fact that an opposite trend was observed in the stabilizing material (Figure 2). Moreover, the D[3,2] values of HPH emulsions are larger than BM emulsions.

Overall, the data of Table 2 show that all the prepared emulsions are in the micrometric range, with the 90th percentile well below 100 μm.

The influence of the different mechanical treatments of cellulose on the flow behavior of the Pickering emulsions was assessed also through flow measurements in steady-state conditions. Figure 6 reports the viscosity of different formulations of Pickering emulsions as a function of the shear rate. It can be noticed that also the rheological behavior of the emulsions was significantly influenced by the mechanical treatment of cellulose. Interestingly, both emulsions stabilized with BM cellulose demonstrated the occurrence of hysteresis (data not shown) within the range of shear rate investigated, suggesting significant instability during shearing. The viscosity of HPH-NC and HPH-TP emulsions showed the typical non-Newtonian shear thinning behavior, which is markedly different from the Newtonian behavior observed for surfactant-stabilized emulsions [34]. The decrease in viscosity can be linked to the breakdown of the entangled network of nanofiber bundles and their orientation along flow lines upon the shear force application [35]. Moreover, it was observed that the HPH-NC emulsion is characterized by higher viscosity values than emulsions prepared with BM-cellulose. The long and flexible HPH fibrils, shown in Figure 4c, likely formed an inter-chain network, which entrapped the oil droplets into cellulose-rich aggregates. Therefore, the rheological behavior of HPH-NC emulsion was dominated by the properties of the cellulose network within the aggregates. This consideration is in good agreement with the observed emulsion microstructure and the interfacial tension measurements.

The influence of temperature on the rheological properties of the emulsions is shown also in Figure 6. Despite the fact that an increase in temperature usually leads to a decrease in emulsion viscosity, the viscosity of all the Pickering emulsions increased with increasing temperature. This is especially evident for the emulsions stabilized with BM cellulose, which also exhibited a markedly different flow-curve shape. The rheological behavior of NC-stabilized emulsions needs to be further investigated in future studies.

This phenomenon could be explained by the effect of temperature on cellulose fibers, for example, through increased swelling as temperature increases and increased hydrophilicity, with the consequence of the formation of a predominantly elastic 3D network, characterized by increased viscosity.

### 3.3. Emulsion Stability

The influence of pH on the physical stability of the Pickering emulsion is of interest for application in processed foods as well as during gastrointestinal digestion [36]. The stability of emulsions stored at 4 °C for 28 d at different pH values (2.0, 7.0, and 12.0) was evaluated upon centrifugation (3500 rpm for 10 and 20 min) in terms of changes in microstructure (Figure 7).

In Figure 7, the red, yellow, and dark regions represent oil droplets, cellulose, and water, respectively. The oil phase is always completely encapsulated by water inside the emulsions in the form of spherical droplets. Large oil droplets for all emulsions were observed under acidic conditions (pH = 2.0), thus indicating substantial droplet aggregation and low emulsion stability across this pH range [37].

During the storage period, creaming (dense layer floating on the top) and/or sedimentation (pellet layer at the bottom) occurred for emulsions stabilized with BM30-CP and BM60-CP, as a consequence of gravitational separation (in detail Appendix A). This observation indicates the poor stability of these emulsions, which is particularly evident at low pH, where a clear serum is formed. Moreover, some oiling off is also observed, suggesting the occurrence of coalescence phenomena. Upon addition of HCl, protonation (H^+^) occurs, which is reported to promote NC aggregation and fast sedimentation in the suspension, leading to instability of emulsions. This phenomenon is due to the attractive forces among the NC sub-units, caused by the van der Waals and hydrogen bonding, which, in an acidic environment, are considerably stronger than the repulsive forces induced by surface charges. Under neutral conditions, the NC suspensions are negatively charged, no aggregation mechanism occurs, and colloidal stability is ensured. In alkaline conditions, however, high-concentration of OH^−^ can induce the disruption of the cellulose crystalline domains, therefore causing phase-separation phenomena in the emulsions [38].

When compared with the emulsion stabilized by BM-treated cellulose, HPH-NC and HPH-TP provided better coverage of the oil droplets surface, thus increasing the dimensional resistance among droplets, which reduced coalescence of the droplets also by changing the pH environmental factor (no oiling off observed after 28 d or centrifugation). In particular, for HPH-NC, neither creaming nor sedimentation was observed during storage, suggesting the high stabilization efficiency of the resulting emulsion at all tested pH values. This could be attributed to the increased cellulose surface charges after defibrillation with HPH, which are responsible for more intense attractive forces, regardless of the environmental conditions. As a result, the emulsion stabilized with HPH-NC was not prone to sedimentation both in acid and alkaline conditions. In the case of HPH-TP, clear serum appeared at the bottom already from day 0, with some evidence of sedimentation only at pH = 2. In general, HPH-treated systems produced more stable emulsions than BM-treated cellulose. The higher stability of emulsions with HPH-NC and HPH-TP is also confirmed by the viscosity analysis over storage period reported in Appendix A. However, in the case of tomato pomace, a higher solid content is likely necessary for emulsion stabilization to compensate for the fact that cellulose is only a fraction of the overall solids (about 37% on the dry matter), and, therefore, lower emulsification ability is recorded.

A detailed study of emulsion stability upon centrifugation at different pH values and of the changes in the rheology of the systems is reported in the Appendix Al, confirming what was already discussed in the present section.

## 4. Conclusions

This study investigated the influence of different mechanical treatments of cellulose, or tomato pomace as a raw cellulose source, on the stabilization of O/W Pickering emulsions. Two mechanical processes were investigated, based on ball milling (BM) for 30 and 60 min, or high-pressure homogenization (HPH). The HPH-treated cellulose exhibited fibril bundles, with a length in the range of ~10–100 μm and irregular widths. This morphological structure exhibited higher mobility and flexibility at the oil–water interface, resulting in an efficient emulsifying ability at different pH values, with the fibrils wrapping around the oil droplets for their stabilization. Despite the fact that BM-treated cellulose showed a finer and more uniform size distribution, the resulting emulsions exhibited oil droplets not completely covered by cellulose fibrils, and, therefore, significantly lower stability. The hypothesis that the higher stability of emulsions stabilized by HPH-treated cellulose is due to the entanglement of fibrils around the droplets to form a 3D network was supported by viscosity measurements: HPH-NC and HPH-TP emulsions exhibited significantly higher viscosity than BM30-CP and BM60-CP emulsions.

## Figures and Tables

**Figure 1 foods-10-01886-f001:**
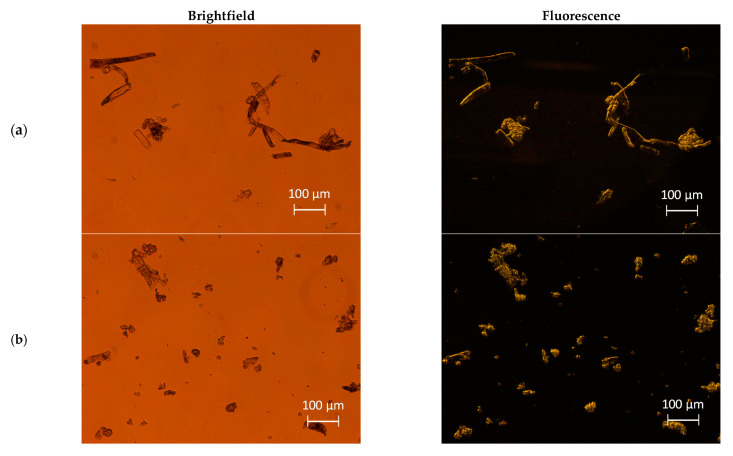
Brightfield (left column) and fluorescence (right column) micrographs of (**a**) untreated cellulose, and cellulose treated by ball milling for (**b**) 30 min (**c**) and 60 min, by (**d**) HPH treatment for 10 min, and (**e**) HPH-treated tomato pomace aqueous suspensions.

**Figure 2 foods-10-01886-f002:**
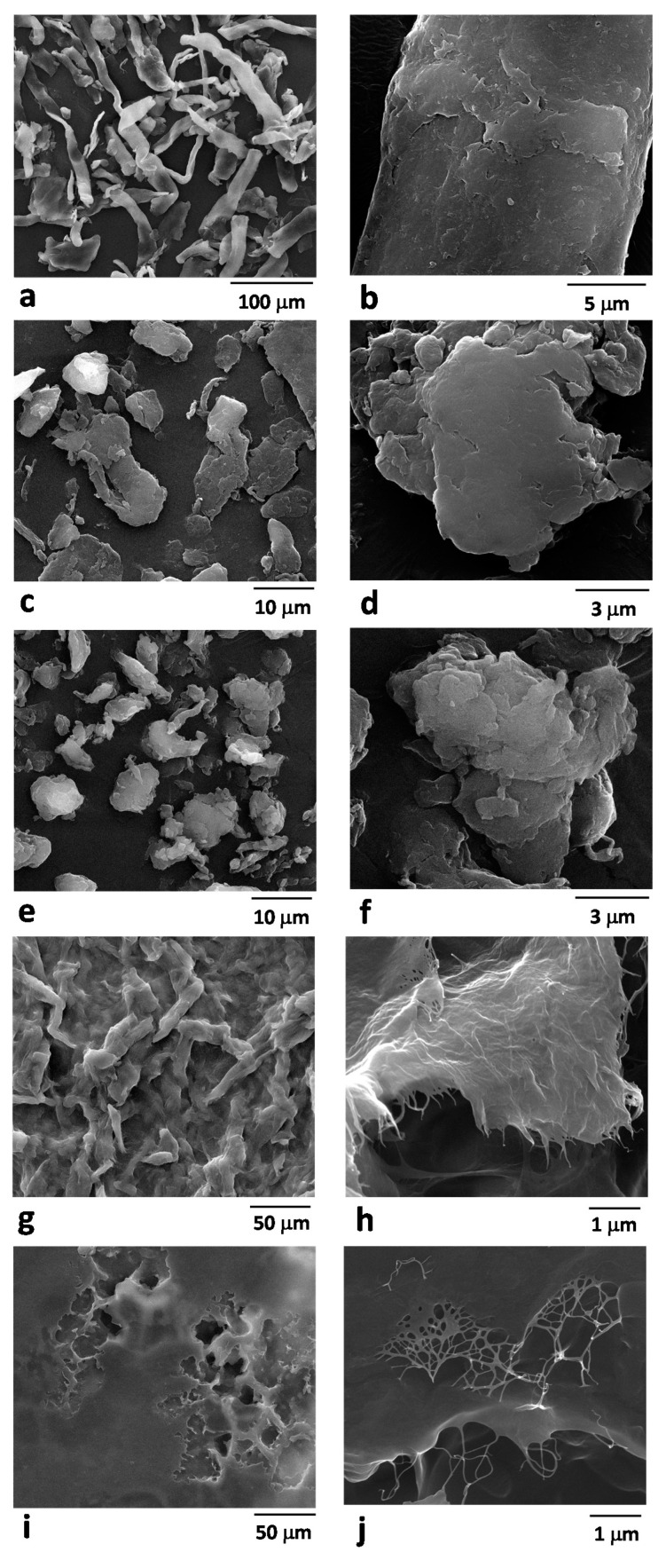
SEM images of (**a**,**b**) untreated cellulose, cellulose treated by ball milling for (**c**,**d**) 30 min and (**e**,**f**) 60 min, by (**g**,**h**) HPH treatment for 10 min, and (**i**,**j**) HPH-treated tomato pomace aqueous suspensions.

**Figure 3 foods-10-01886-f003:**
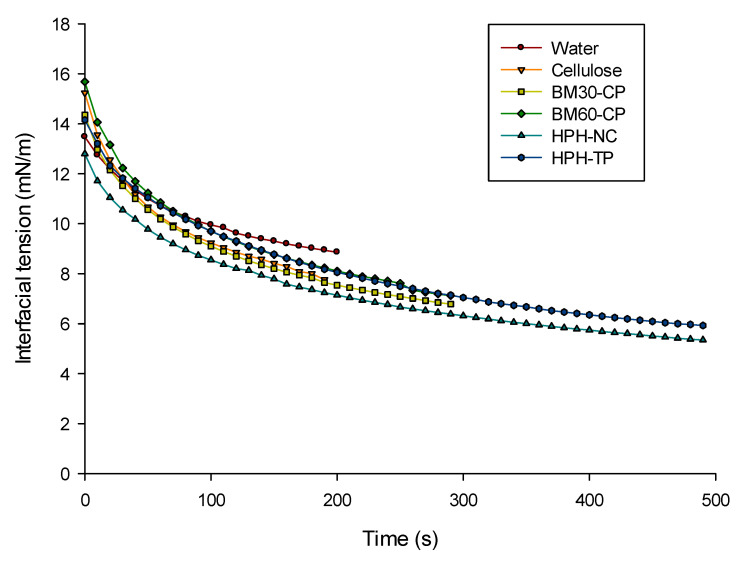
Interfacial tension dynamic curves at water–oil interface, for pure water, and aqueous suspensions containing untreated cellulose, cellulose subjected to different mechanical treatments and HPH-treated tomato pomace. All aqueous suspensions were prepared at 0.5%_DM_. Each curve is the average of five measurements.

**Figure 4 foods-10-01886-f004:**
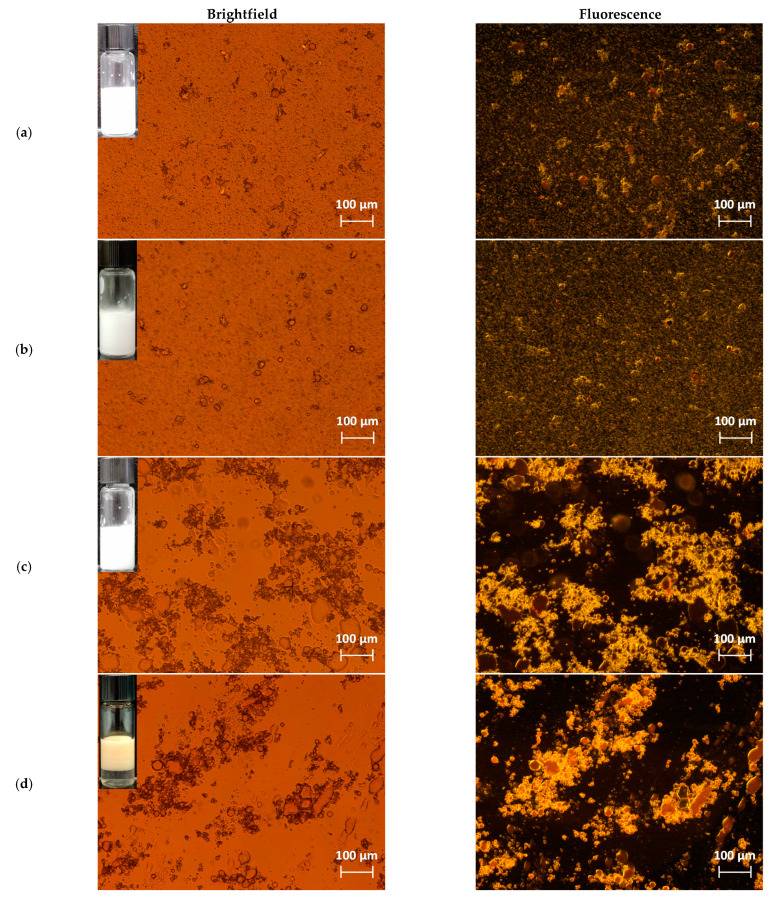
Comparison between brightfield (left column) and fluorescence (right column) micrographs of emulsion stabilized with (**a**) BM30-CP, (**b**) BM60-CP, (**c**) HPH-NC, and (**d**) HPH-TP.

**Figure 5 foods-10-01886-f005:**
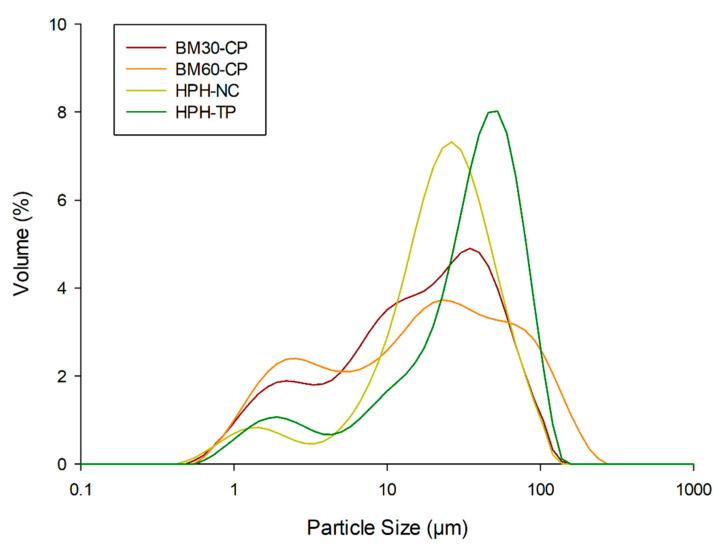
Particle size distribution of Pickering emulsion stabilized with (red line) BM30-CP, (orange line) BM60-CP, (light green line) HPH-NC, and (dark green line) HPH-TP.

**Figure 6 foods-10-01886-f006:**
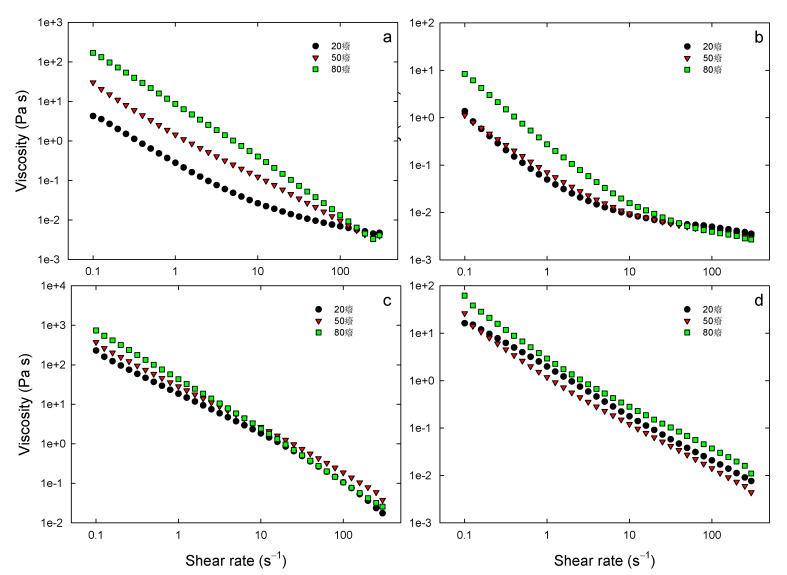
Flow measurements (apparent viscosity as a function of shear rate) in steady-state conditions at different temperatures of freshly prepared emulsion stabilized with BM30-CP (**a**), BM60-CP (**b**), HPH-NC (**c**), or HPH-TP (**d**). Each flow curve is the average of five measurements.

**Figure 7 foods-10-01886-f007:**
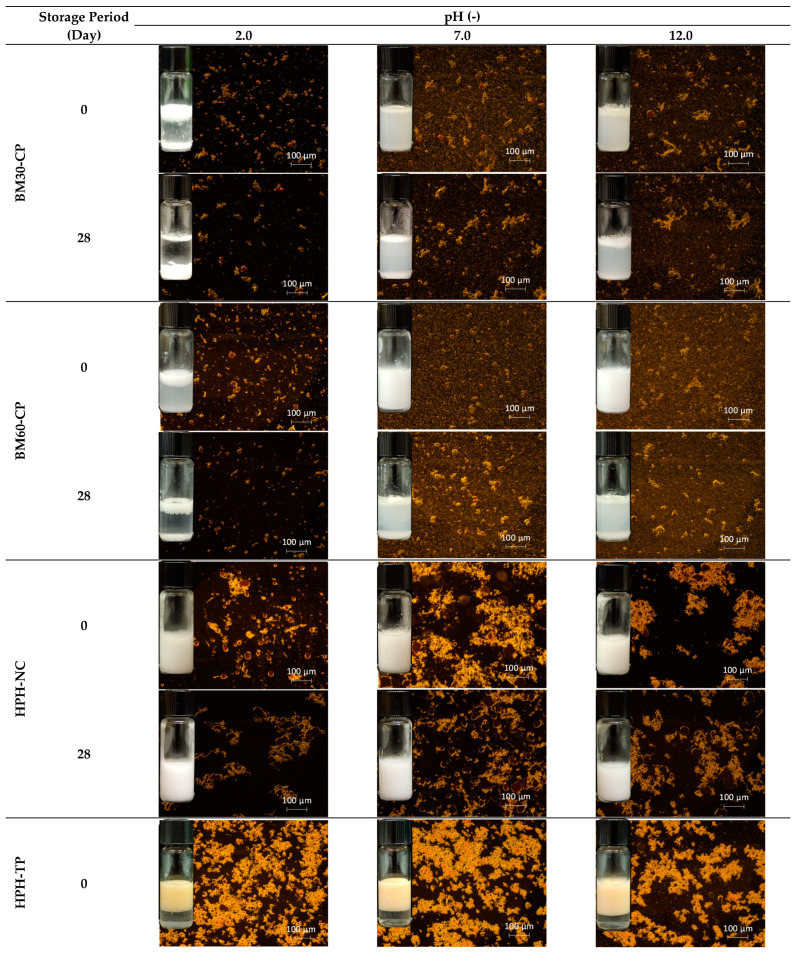
Fluorescence microscopy and visual images of emulsions stabilized by cellulose and tomato pomace subjected to different mechanical treatments over 28 d of storage, at 4 °C, at different pH values.

**Table 1 foods-10-01886-t001:** Effect of different mechanical treatments of cellulose and tomato pomace dispersed in water on particle size distribution (expressed as span and characteristic diameters) in comparison with pure water and untreated cellulose suspensions.

	Cellulose	BM30-CP	BM60-CP	HPH-NC	HPH-TP
d(0.1) (µm)	14.20 ± 017 ^a^	7.16 ± 0.14 ^c,d^	6.89 ± 0.12 ^d^	9.56 ± 0.11 ^b^	7,78 ± 0.14 ^c^
d(0.5) (µm)	53.23 ± 2.24 ^a^	31.69 ± 0.34 ^b^	30.66 ± 0.32 ^b,c^	32.04 ± 0.12 ^b^	29.97 ± 0.25 ^c^
d(0.9) (µm)	206.46 ± 13.96 ^a^	88.82 ± 0.69 ^c^	82.93 ± 0.59 ^d^	108.73 ± 1.16 ^b^	78.22 ± 0.50 ^e^
Span (-)	3.61 ± 0.15 ^a^	2.58 ± 0.02 ^c^	2.48 ± 0.02 ^dc^	3.10 ± 0.03 ^b^	2.35 ± 0.01 ^d^
D[4,3] (µm)	100.43 ± 5.32 ^a^	45.01 ± 0.65 ^c^	44.99 ± 0.49 ^c^	54.44 ± 0.43 ^b^	38.47 ± 1.06 ^d^
D[3,2] (µm)	23.25 ± 0.29 ^a^	12.79 ± 0.18 ^d^	11.68 ± 0.17 ^d^	16.85 ± 0.12 ^b^	14.47 ± 0.16 ^c^

Different letters denote significant differences (*p* < 0.05) among the different samples within each row (*n* = 3).

**Table 2 foods-10-01886-t002:** Particle size distribution (expressed as span and characteristic diameters) of emulsions stabilized with BM30-CP, BM60-CP, HPH-NC, and HPH-TP.

	Pickering Stabilizer			
	BM30-CP	BM60-CP	HPH-NC	HPH-TP
d(0.1) (µm)	1.92	1.79	5.02	3.82
d(0.5) (µm)	15.60	16.65	21.75	34.40
d(0.9) (µm)	54.44	84.25	53.85	72.73
Span (-)	3.37	4.95	2.245	2.00
D[4,3] (µm)	22.90	31.11	26.38	37.05
D[3,2] (µm)	5.70	5.44	8.45	9.83

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
