# Peer review of "O/W Pickering Emulsions Stabilized with Cellulose Nanofibrils Produced through Different Mechanical Treatments"

_foods, 2021, doi:10.3390/foods10081886_

Round 1

Reviewer 1 Report

The manuscript describes a study of the processing of different sources of cellulose into particles and fibrils and investigate their emulsifying properties. In my opinion, the investigation is not thorough and the interpretations are often highly speculative. It is not clear what new knowledge is generate from this study. There are some serious shortcomings in the interfacial tension experiments detailed below. 

There have been a number of papers studying the emulsifying properties of cellulose particles from food waste sources, the authors need to highlight what is new about their approach.

Why was the tomato pomace only treated with HPH and not ball milling, please justify.

Line 204 what is meant by "increasing their bonding potentials" following treatment, and what evidence is there of this.

Line 205 and 220 onwards. it is difficult to assess the effect of HPH on tomato pomace, as there is no images before HPH, and no obvious evidence in Fig 1e of "cell disruption into filamentous debris" as there are clearly still some intact cells. Looking at the particle size, for HPH-TP most of the particles are below 30 microns, which is not evident from the microscopy.

The interfacial tension measurements do not indicate very well the surface activity of the individual preparations. The oil sample clearly has some very surface active material in it as it reduces the interfacial tension quickly down to  5.7 mN/m. As such, the competition between the cellulose samples and the impurity in the oil phase means it is very difficult to assess the interfacial activity of the test samples. These experiments should be repeated using a purer oil phase, with little or no intrinsic surface activity.

Line 270 - the whole discussion around the adsorption of cellulose fibres is speculative. Certainly for the tomato pomace, as this sample will contain a whole range of surface active molecules that are more surface active than cellulose. The impact of BM on surface activity could also arise from impurities arising from the processing rather than the cellulose itself. Cellulose particles, tend to adsorb very slowly due to their very low diffusion coefficient. The authors need to rethink this whole section on interfacial tension both in the experimental methodology and the interpretation.

Line 359 - 364. The statement that emulsion droplet interactions contribute to the viscosity is highly speculative and probably wrong. The authors have demonstrated that the HPH emulsions show highly flocculated/aggregated networks that entrap the emulsion droplets. So how can they speculate that the higher viscosity is due to enhanced interactions between emulsion droplets. There is no evidence for this, and the images clearly show that the structure is dominated by the aggregation of the HPH preparation. In addition, the HPH-TP shows very rapid phase separation and creaming in Fig 5, evidencing the presence of large floccs containing the emulsion droplets, note the clear serum phase indicating the absence of either oil droplets or cellulose aggregates. Therefore the droplets are entrapped within cellulose-rich aggregates and hence the interactions between the droplets is dominated by the properties of the cellulose network within the aggregate. This section should be removed or rewritten to reflect this.

Reviewer 2 Report

The publication is correct, also the results are clearly presented and properly commented. The weak point of the manuscript is the introduction, which will not introduce the essence of the research. Other  weak point is the section "Mechanical treatments of cellulose and tomato pomace", which lacks an analysis of the size and particle structure obtained after mechanical treatment. It was not shown how many repetitions were made to perform preparation of Pickering emulsions,  measurements of interfacial tension, scanning electron microscopy analysis, size distribution of BM-CP, NC, and HPH-treated tomato pomace.

It was metioned that the pH of the emulsions was adjusted  from 2 to 12 using 0.1 mol/L of HCl solution and 0.1 mol/L of NaOH solution, however only results for pH 2, 7 and 12 were presented. Why the  pH values were chosen? how does it relate to the emulsion application? was the pH controlled after emulsion or was only the initial solution controlled? Why the discussion did not analyze the possibility of CN reaction or with H + or OH-?

Reviewer 3 Report

This study presents relatively novel advances on the development of Pickering emulsions. It is well written in general, highlighting especially a good introduction and well described materials and methods. In my opinion, the manuscript can be considered for publication after a review of some aspects.

Line 58. This phrase requires a greater number of citations. Recent studies preferably.

Section 2.7. Why was the span not also evaluated? Possibly more informative than using so many diameter values.

Section 2.8. Why have the flow curves not been carried out using shear stress? The AR200 is a stress controlled rheometer, not a speed controlled rheometer. Also, why have no oscillatory tests been performed? Provide the same or more information than flow curves

Section 2.8. The power law, as it has been expressed and from a strictly mathematical point of view, should not be used to compare between samples. the reason is that the consistency index has units that depend on the flow index. There are studies that use a useful modified version in these cases, such as:

Trujillo-Cayado, L. A., Natera, A., García González, M. D. C., Muñoz García, J., & Alfaro Rodríguez, M. D. C. (2015). Rheological properties and physical stability of ecological emulsions stabilized by a surfactant derived from cocoa oil and high pressure homogenization.

Results. In addition to including the span data, as well as its interpretation, I suggest including the droplet size distributions, as they provide additional information. If evolution is available over time, its inclusion would also be important.

Results. I suggest including the power law tuning parameters for all systems (including R2). Apparently the BM60-CP does not seem to fit too well

Round 2

Reviewer 1 Report

The authors have revised the manuscript and addressed most of the issues I raised in my review of the original submission and the manuscript is now significantly improved.

However, there are still some major issues with the interpretation of the  interfacial tension experiments. I would like to thank the authors  for repeating the measurements and supplying the kinetic data in the supplementary information. This is very useful for the interpretation. It can be seen here (Fig S2) that the interfacial behaviour of  all of the samples is totally dominated by the behaviour of the control sample. The test samples also all behave in a similar manner, so it is extremely difficult to come to any conclusions about the significance of the statistical differences between the actual values. I am afraid that filtering with a PTFE filter will not remove much surface active impurities. Treatment with activated magnesium silicate (Florisil) is often used to remove oil soluble surface active impurities. Otherwise, from an academic perspective, to determine the surface activities of the samples, use a pure triglyceride oil such as tri-olein which will allow a much better discrimination between the control and test samples.

Otherwise, if the authors wish to keep the current data, the text in the manuscript will have to be significantly changed to reflect this and the points made below:

  1. The oil phase still contains significant impurities. All samples do show a moderate reduction in interfacial tension compared to the controls, but the behaviour of all the test samples is similar. Only the HPH-NC sample is consistently lower than the rest.
  2. The BM-30 and BM-60 samples do NOT show enhanced surface activity compared to the untreated cellulose. The asymptotic values may show a significant difference, but the data in Fig S2 show cellulose and BM-30 are almost identical, and BM-60 is actually higher! Therefore this has to be removed from the text. This also suggests the quality of the fits of the model to the data may be responsible for the differences in the asymptotic values.
  3. Cellulose particles are not particularly surface active in the absence of salt (see http://dx.doi.org/10.1021/acs.langmuir.8b03056 and https://doi.org/10.1039/C9NA00506D) therefore it is not surprising that there is little surface activity beyond the control, considering that the control measurements show significant surface activity.
  4. The fact that the cellulose particles are not very surface active in the absence of salt, and that the control values also show surface activity, the impact of the cellulose samples on the overall shape of the curve is small. Therefore it is not possible to draw conclusions about the molecular relaxation times etc drawn from the parameters from the fitted model. Therefore this section should be removed.

Therefore either if the experiments are repeated with an oil of much higher purity, or the text is changed in accordance with the points above to more accurately represent the experimental data, then I will be happy to recommend the manuscript for publication.

Author Response

We sincerely thank the reviewer for the help in improving the discussion of our result, with specific reference to the interfacial tension measurements.

We understand the points made by the reviewer, and, therefore, we decided to modify the manuscript as follows:

  • We have removed the fitting of the curves and we have presented (Figure 3) the dynamic curves, instead. This is helpful for supporting the discussion, as suggested by the reviewer.
  • We have simplified the discussion, focusing on the main points suggested by the reviewer, instead of a more complex (and unsupported by experimental evidence) discussion of the small differences in asymptotic interfacial tension values and relaxation times.
  • The discussion now focuses mainly on the fact that HPH-treated samples is significantly from the others, while untreated cellulose and BM-cellulose do not differ from the control sample.

We are now convinced that the manuscript better reflects, in the discussion, the evidence of the experimental measurements.

The manuscript now reads (lines 276-301): “The results reported in Figure 3, as kinetic data of interfacial tension, clearly show that the oil phase still contains significant impurities, also after filtering the oil with a PTFE filter. Therefore, the interfacial behavior of the samples is dominated by the be-havior of the control sample. In general, the systems where cellulose was added did not show any noteworthy reduction in interfacial tension compared to the control, with the exception of the HPH-NC sample, which exhibited an interfacial tension that was consistently lower than the other samples. More specifically, the BM30-CP and BM60-CP samples did not show an enhanced surface activity compared to the untreated cellulose, with the dynamic interfacial tension curves for cellulose and BM30-CP that are almost identical, and the BM60-CP curve slightly higher. The observation that the dynamic curves for cellulose suspensions do not differ from the curves for pure water in oil is consistent with previous data showing that cellulose particles are not particularly sur-face active in the absence of salt [27,28].

The lower interfacial tension dynamic curves observed for HPH-NC could be likely ascribed to the morphology changes induced by HPH, promoting defibrillation but less efficient fiber breakage than BM. Moreover, it can be hypothesized that HPH treatment caused the cellulose to exhibit a lower interfacial tension than cellulose, due to the stronger attractive capillary forces establishing for the softer particles generated by HPH treatment than for rigid ones [30]. In the case of tomato pomace, its complex composition, and reduced cellulose content, might explain the reduced differences observed with the control sample, despite the HPH treatment.”

These changes respond to all the points raised by the reviewer:

  1. We have clearly specified that the oil phase still contains significant impurities and that the behavior of all the test samples is similar, with only the HPH-NC sample being consistently lower than the rest.
  2. We have also stated that the BM-30 and BM-60 samples do NOT show enhanced surface activity compared to the untreated cellulose. The discussion about asymptotic values has been removed.
  3. We have compared our results with that of cellulose particles in the absence of salt, as suggested.
  4. We have also clearly stated that the impact of the cellulose samples on the overall shape of the curve is small. The discussion about molecular relaxation times etc has been removed.

We hope that the manuscript is suitable for publication.

Reviewer 2 Report

The authors responded to the suggestion and made all changes as indicated. The current version of the manuscript may be published

Reviewer 3 Report

Taking into account the reviews made by the authors and the quality of the manuscript, I consider that it can be accepted for publication in its current version.

Author Response

We sincerely thank the reviewer for the help in improving our manuscript.